# Who Prefers Legal Wood: Consumers with Utilitarian or Hedonic Shopping Values?

Pipiet Larasatie [1,*], Radityo Putro Handrito [2,3,*], Triana Fitriastuti [3,4] and Dhina Mustika Sari [2,4]

1    Arkansas Center for Forest Business, College of Forestry, Agriculture & Natural Resource, University of Arkansas at Monticello, Monticello, AR 71656, USA

2    Faculty of Economics and Business, Department of Management, Universitas Brawijaya, Kota Malang 65145, Indonesia; dhina_sari@student.ub.ac.id

3    Faculty of Economics and Business Administration, Ghent University, 9000 Ghent, Belgium; triana.fitriastuti@ugent.be

4    Faculty of Economics and Business, Universitas Mulawarman, Kota Samarinda 75119, Indonesia

*    Correspondence: larasatie@uamont.edu (P.L.); radityohandrito@ub.ac.id (R.P.H.)

**Abstract:** Although certification is perceived to be beneficial for enhancing forest sustainability and open access to green markets, certification practices in Indonesia face controversy, particularly in its wood-based industry. We aim to approach this issue from the end-user perspective. Drawing on the theories of value-belief-norm and planned behavior, we examine the psychological aspects of consumers toward legal wood consumption. A survey of 515 consumers showed that individuals with hedonic values tended to have a high perception of green values toward legal wood. Also, when consumers' hedonic values dominated over their utilitarian consumption, their perception of green values toward legal wood tends to be higher. Based on these results, wood marketers could benefit from directing their communication efforts toward emphasizing the hedonic worth of the product, as opposed to its utilitarian values. It is imperative for distributors and promoters of wood products to carefully deliberate on strategies to effectively elicit the hedonic shopping values that are inherently linked to the utilization of such green products. An illustration can be represented by the case of IKEA in Indonesia. Consumers are probably attracted to IKEA's neuromarketing strategy, such as their attractive display and labyrinth effect, without realizing that IKEA also applies green marketing and supports green products.

**Keywords:** legal timber; certified wood; forest certification; green products; shopping values

## 1. Introduction

Forests, covering roughly one-third of the earth's landmass, provide a vast array of ecosystem services, including reliable clean water, productive soils, and climate regulation [1]. Approximately one-fifth of the total human population (1.6 billion people) depend directly on forests for their livelihoods [2]. This dependence can be severely affected by deforestation and forest degradation.

Despite having the largest tropical rain forest biome in the world, Indonesia is reported to also have one of the largest primary forest losses [3]. The depletion of the natural forest in Indonesia is caused by several factors, including weak or compromised public governance, flawed corporate governance, and the monopsony of wood supply mechanisms [4]. Deforestation was perceived to be a problem when Indonesia began to use forest resources for their economic benefits and established large-scale commercial logging concessions [3]. These circumstances portrayed the complexity of the wood supply chain and the absence of wood pricing transparency in Indonesia, resulting in industrial overcapacities [4]. The high rate of deforestation in Indonesia caused by economic factors has been a high priority concern, not only by the Indonesian government, but also by international bodies such as IMF/World Bank [5].

Once the world's leader in roundwood and plywood (Figures 1–4), the woodworking sector in Indonesia has significantly declined [6]. Wood-based products from Indonesia have been unable to meet international demand due to their legality and sustainability issues [3]. To supplement their wood supply shortage, Indonesia is suspected of using illegal timber [3].

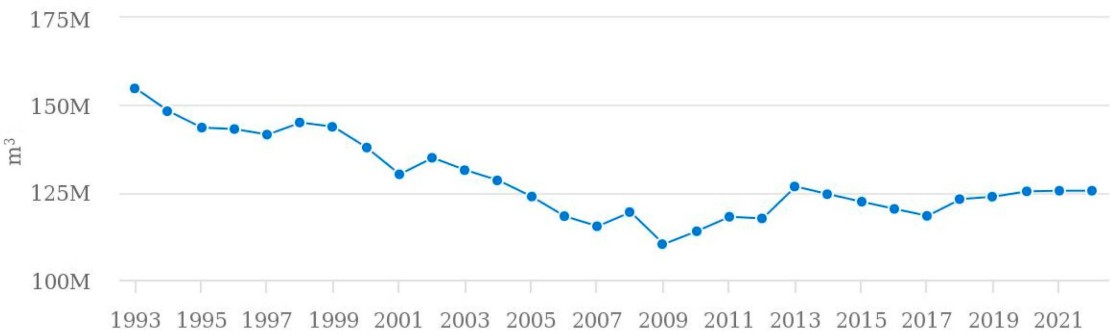

**Figure 1.** Roundwood production quantity in Indonesia [7].

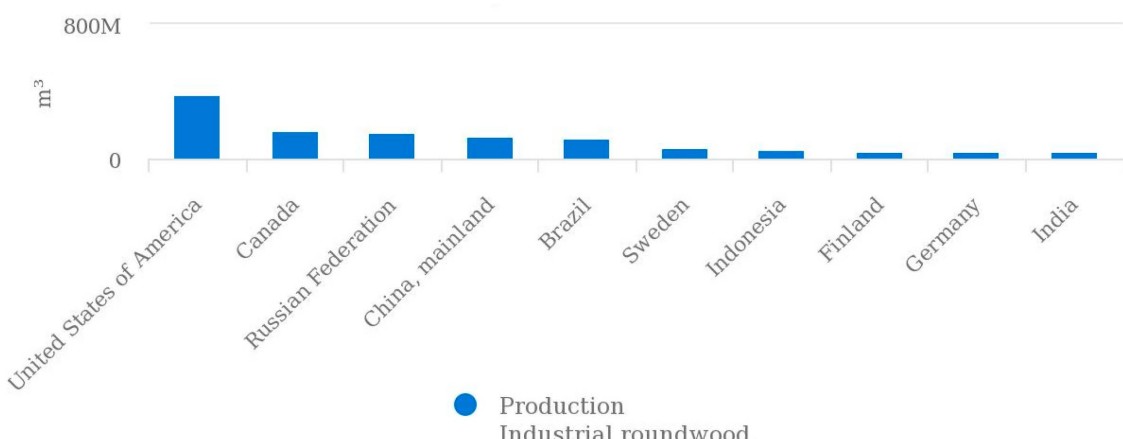

**Figure 2.** Top 10 countries for roundwood production quantity [7].

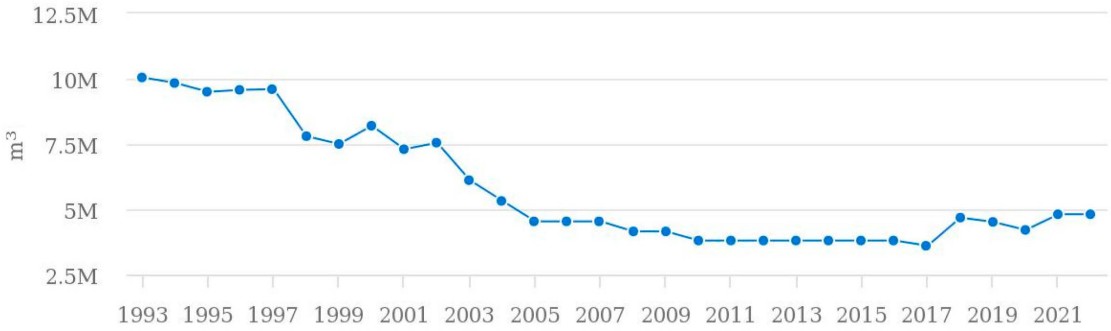

**Figure 3.** Plywood production quantity in Indonesia [7].

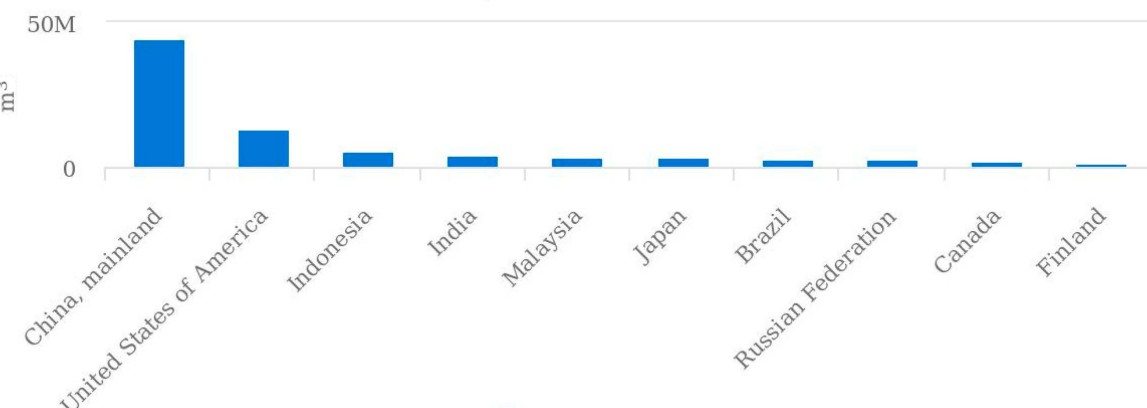

**Figure 4.** Top 10 countries for plywood production quantity [7].

A study compiling the history of forest loss and degradation in Indonesia suggested promoting legal wood as a solution for reducing the rate of forest loss [3]. Legal wood is obtained through a standardized certification process. Certification is important, not only for natural resource management, but also for human resource management, such as improving worker safety measures, augmenting worker training programs, and enhancing communication and dispute resolution strategies with stakeholders, neighbors, and communities [8].

The impetus for the involvement of the Government of Indonesia in forest certification was sparked by a convening of meetings by the International Timber Trade Organization (ITTO) throughout the latter part of the 1980s and the early years of the 1990s [9]. During this period, two alternatives were subject to deliberation: participation in ongoing established international certifications, such as the Forest Stewardship Council (FSC), or establishing a distinct national certification process, system, and standard that operated independently from external processes. The stakeholders in Indonesia opted for the second alternative, wherein the certification process was initiated by the producers themselves, independent of any other international initiatives.

This national system, called *Sistem Verifikasi Legalitas dan Kelestarian* (SVLK) or the Forest Legality and Sustainability Assurance System, verifies the legality of forests and forest products originating from Indonesia's forests, confirming that they are legally guaranteed and certified as being sustainably maintained [10]. In the implementation of this system, the Government acts as the regulator, overseeing assessment and verification procedures. Various stakeholders are involved in these procedures, including the National Accreditation Committee, business enterprises and their representative organizations, as well as independent monitors, such as non-governmental organizations and academic institutions. SVLK offers two types of certifications: the Sustainable Production Forest Management Certification and Timber Legality Certification [10].

The approach closely resembles the establishment of Malaysia's National Timber Certification Council [9]. Nevertheless, both nations have since made the decision to pursue stronger connections with global certification organizations, particularly the FSC, in order to obtain international acknowledgement for their own labels. Receiving a certification ensures a green label, providing assurance to consumers that the wood was harvested in a sustainable manner [11]. Products that are made from legal wood are categorized as green products.

However, although certification is perceived as being beneficial to enhancing sustainability and opening access to green markets [11], the practices in Indonesia have faced controversy. Wibowo and Giessen [12], for example, questioned the ability of certification to reduce forest loss and sustain forest management in Indonesia. Rather than efforts to secure existing tropical forests, this certification and legality verification regime are arguably mere tools in governing the timber trade at the expense of small and traditional tree growers [12].

A study conducted in the United States revealed that despite small woodland owners possessing over 40% of the forest land in the region, they exhibited a significant reluctance to pursue certification, primarily attributed to the exorbitant expenses associated with the process [13]. They believed that the expenditure of resources (money and effort) on certification was burdensome, lacking in visible advantages for landowners.

In Indonesia, the pros and cons of certification are also seen in the wood-based furniture industry [14]. Studies from Purnomo et al. [15] and Larasatie [16] show increasing consumer preferences for the production of green labeled furniture made from legal wood. Market-based certification and labeling operate under the assumption that, as market access and environmental awareness improve, customers become more inclined to financially incentivize producers through a price premium scheme [17]. This phenomenon is made feasible through the adoption of a market-oriented approach [18]. However, though green labeling is considered to have a positive impact on the industry image, not all furniture buyers demand certification. As a result, there are producers who view green labeling as unfair competition from developed countries and that it was actually implemented to be a barrier to entry the global trade [14]. Rather than an instrument of environmental management, green labeling is even considered as a new form of colonialism. These findings support a statement by Bartley [19] who found that forest product certification in Indonesia was export dependent and experienced high degrees of controversy.

In the context of Indonesia and other developing countries, issues also arise from consumer perspectives. Products made from legal wood, with their premium price, may be perceived and marketed as luxury items. On the other hand, items made from non-legal wood are usually cheaper, as there are no additional costs related to the certification process. Therefore, consumers make a choice between at least two significant possibilities when it comes to products made from wood. In this scenario, buyers are compelled to carefully evaluate their choices based on specific attributes, such as price vs. environmental sustainability. Individuals choose between purchasing affordable products that meet their fundamental need for wood-related items or investing in more expensive options that not only fulfill their needs, but also have a reduced negative impact on the environment. Therefore, consumer perceptions of such products are crucial in this context.

We aim to approach these issues from an end-user perspective. Thus, we bring the psychological aspects of consumers toward legal wood (in this case, theorized through the perception of green values). Drawing on the theories of value-belief-norm [20,21] and planned behavior [22], we explore utilitarian and hedonic shopping values [23]. We investigated which shopping value would influence consumer preference toward using legal wood, and what the possible outcomes would be when both values interacted with each other. We utilized polynomial regression with surface analysis as analytical tools to identify the discrepancies and congruencies between hedonic and utilitarian values.

This study presents at least three contributions. First, the study examines the applicability of management theories, such as value-belief-norm and planned behavior, in the context of Indonesia. Specifically, we aim to evaluate whether these theories, which were originally created in Western settings, can also be applied to a less developed country. Moreover, it is worth noting that there is a scarcity of studies examining green buying and consumption behavior within the Asian demographic. Second, from a methodological standpoint, the utilization of polynomial regression with surface analysis is employed. The utilization of this analytical approach enables greater comprehension of intricate variations and agreements among autonomous variables, as well as their directional association with the dependent variable. Third, it is evident that the inclusion of psychological factors plays a crucial role in comprehending green values. Incorporating this element will offer a more comprehensive framework for academia, and marketers will be able to suggest a marketing strategy better aimed at promoting the adoption of environmentally friendly products.

## 2. Research Context: Forests in Indonesia

Indonesia, the largest archipelagic country in the world, has designated 63 percent of its total land area (120.5 million hectares) as state forest area [10]. Indonesia's forest area is categorized according to three different functions: (1) Production forests (68.8 million hectares); (2) Protection forests (29.6 million hectares); (3) Conservation forests (22.1 million hectares). Production forests consist of permanent production forests, limited production forests, and convertible production forests [10]. Protection forests are forest areas designed as buffer zones to regulate hydrology, control floods, prevent erosion, avert abrasion; and maintain soil fertility.

Through a re-evaluation of land cover using image interpretations from the Landsat Data Continuity Mission/Landsat 8 OLI for 2020, it was determined that 79.9 percent of Indonesia's conservation forest areas, 81.7 percent of its protection forest areas, and 81.2 percent of its limited production forest areas were forested [10]. The forest cover in permanent production forest areas was 63.6 percent, whereas in convertible production forest areas the forest cover was 50.2 percent.

In accordance with Indonesian legislation, the Ministry of Environment and Forestry (MoEF) has jurisdiction over regions officially designated as the forest area [10]. Therefore, forest management is driven by the government of Indonesia through the MoEF, along with stakeholders, including but not limited to, local governments, local communities, non-profit organizations, and religious groups [10,24].

## 3. Theoretical Background and Hypothesis Development

### 3.1. Psychological Aspects of Consumers

The consumer's perception toward green products is influenced by various factors, both external and internal [25,26]. External factors can include culture, social class, advertisements, economics, and political aspects [27–30]. Research on consumer willingness-to-pay for environmentally certified wood products in the United States revealed significant positive associations between willingness-to-pay and environmental consciousness, certification involvement, and perceived importance of certification [31]. These elements are typically transient and subject to alteration because of the fluidity of circumstances [32]. Hence, it is imperative to continuously update the stimuli regarding external issues in order to stay abreast of the dynamics and shifts in consumer preferences.

On the other hand, compared with external factors, internal elements are perceived as being more stable and remaining for an extended duration [33,34]. This concept is intrinsically linked to the unique characteristics of individuals, thereby encompassing their psychological dimensions. Examples of internal factors that influence individual behaviors are perception, motivation [35], attitude, beliefs, experience [36], self-concept, and values [37]. These elements separately or even simultaneously lead to an individual's perception of their surrounding environment, including their perception of green values. Due to our focus on a developing country, we prioritized these internal factors over external factors. This strategy was chosen based on the sustainability transition analyses performed in developing countries by Wieczorek [38].

We utilized the value-belief-norm theory [20,21] to link between individuals' values and their perceptions toward green products. We did so because this theory, grounded in the social psychology of environmentalism, emphasizes the significance of conceptualizing and examining personal values and beliefs in relation to their potential impacts on environmentally conscious consumption [39,40]. In addition, we also used the theory of planned behavior (TPB) [22] as our foundation for predicting consumer behaviors. TPB postulates that individual behaviors are affected by three determinants: subjective norms, attitudes, and perceived behavior control. These theories have served as the foundation for several concepts that suggest the existence of a precursor to an individual's attitude, perception, or even actual behaviors.

Closely related to consumption choices, we investigated the concepts of hedonic shopping values (HV) and utilitarian shopping values (UV) [41,42]. We did so because both values capture a fundamental aspect of the human decision-making process, which is their consumption [43]. The categorization of hedonic and utilitarian values is predicated upon an inherent inclination of consumers toward their shopping activity [41,42]. These two distinct categories of shopping values have been identified as essential motivational orientations influencing consumer behaviors [42]. Based on social exchange theory [44], individuals with utilitarian and hedonic values may also be capable of showing the opposite characteristics in their behavior [23].

In this study, we examine how an individual's environmental concern can be seen as both a functional rational value and socially desirable behavior. We explore these two types of purchasing values as psychological factors that precede environmental concern. The objective of this study was to analyze the aligning and contrasting behaviors displayed by individuals in relation to their perception of green values.

Perception of values refers to the comprehensive assessment made by consumers on the overall benefits of a specific product or service, encompassing their evaluation of both the acquired advantages and incurred sacrifices of their purchase [45,46]. According to the above definition, the perception of green values is defined as a consumer's holistic evaluation of the net benefits of a particular product or service. This assessment considers the individual's environmental preferences, sustainable expectations, and green needs [46].

### 3.2. Utilitarian Values and Perception of Green Values

The concept of utilitarian values (UV) in shopping behavior is the notion that individuals are motivated by the practical purpose of a product and tend to employ logical reasoning in seeking solutions to their problems [41,47,48]. They are also recognized for their meticulous evaluation of the cost-benefit aspects in relation to their consumption [49,50]. Societal pressure encourages individuals to perceive shopping as a necessary task that should ideally be carried out with utmost effectiveness [23]. As an illustration of UV in general, the primary method humans address hunger is through the consumption of food, irrespective of its brand, geographical origin, or other attributes. Regardless of the specific type of food consumed, the primary concern lies in its adequacy for sustenance and nutritional value. We posit that this approach is applicable to the context of green products as well.

The production of environmentally friendly products is driven by the recognition that some products may pose a threat to the natural environment, both in terms of their production methods and the environmental damage they generate [51]. Consequently, there has been an increased introduction of environmentally friendly products to consumers, primarily characterized by their adherence to green values, such as a smaller carbon footprint, less artificial chemicals, and, in the context of wood products, certified or legally sourced timber. The promotion of these environmentally conscious principles is aimed at persuading customers to actively participate in the preservation of nature through their purchasing patterns [51]. The acceptance of green values is becoming widespread, particularly among consumers who possess a strong understanding of environmental-friendly actions. However, this phenomenon primarily occurs in developed countries where people's basic needs have been met, and there has been a comprehensive emphasis on environmental education since early ages [52]. In order to foster widespread societal knowledge and understanding of environmental challenges, it is imperative that awareness is cultivated. Consumers' opinions of the environmental friendliness of various products is influenced by rational thought and a desire for environmental solutions [52].

Interestingly, studies show that individuals who possess a greater degree of utilitarian values tend to prioritize cost savings when engaging in green behaviors [53]. This phenomenon can be attributed to the perception that these products have high prices [54–56]. In this scenario, individuals with a greater inclination toward utilitarian values will prioritize cost-benefit analysis and opt for sensible justifications rather than indulging in lavish pursuits.

Within the parameters of our research, products made from legal wood are viewed as luxury goods with specific attributes that enhance their primary purpose, commanding a premium price. As individuals with utilitarian value place a higher priority on cost savings, their perception of the luxurious benefit of items diminishes. As a result, the propensity toward utilitarian values may result in a decrease in the frequency of environmentally conscious behaviors.

Individuals who prioritize utilitarian values exhibit a decreased level of concern toward environmental issues, as their primary focus lies in the economic advantages derived from the environment rather than its intrinsic benefits. A significant challenge in fostering green behavior lies in the lack of immediate consequences associated with numerous ecological issues [57]. For example, the public encounters challenges in perceiving the presence of the ozone hole, nuclear radiation, and accumulation of greenhouse gases within the Earth's atmosphere. Individuals who prioritize utilitarian values tend to exhibit a shorter time horizon due to their emphasis on financial criteria for achieving their goals [58]. This phenomenon can be explained by Construal Level Theory (CLT) [59,60].

According to the CLT, individuals tend to employ specific, tangible construal at a lower level of abstraction when representing events that are in close proximity. Conversely, when representing events that are further removed, individuals tend to utilize more generalized, abstract construal at a higher level of abstraction. Consumers exhibiting utilitarian values are inclined to engage in extensive consideration of the anticipated rational and tangible advantages linked to products. Tangari et al. [61] suggested that the level of consumer elaboration has an impact on their distance perceptions when it comes to making sustainable choices. Consumers tend to link sustainable products and choices with goals that are more distant and abstract in nature, which is referred to as "high-level construal" [40]. Such perceived behavior–value incongruence does not lead to the activation of sustainable motivation [62]. Consumers who possess a heightened degree of utilitarian shopping values may exhibit reduced receptiveness toward overarching and abstract objectives, such as long-term environment protection. Therefore, the likelihood of engaging in environmental activities, such as using green products, is reduced.

All things considered, the emphasis on monetary value in this example stems from the major consideration of cost-benefit analysis, which emphasizes the rational foundation of utilitarian values. Their perception toward legal wood is mostly focused on short-term monetary gains rather than on its environmental benefits. Hence, we propose that:

**H1:** *Utilitarian values negatively relate to perception of green values toward legal wood.*

### 3.3. Hedonic and Perception of Green Values

Meanwhile, individuals who are motivated by hedonic values are inclined to pursue pleasure, enjoyment, and sensory gratification through their consumption activities [63,64]. Their decisions are mostly motivated by the impulsive and transient satisfaction obtained from the act of consumption. The significance of consumption goes beyond mere utility and serves as a driving force for consumers to actively seek out experiences associated with luxury, celebrations, leisure, or indulgence. These experiences are obtained from the potential benefits of entertainment and emotional satisfaction that are linked with shopping activities [65].

Studies show that hedonic values have a positive impact on personal relevance and importance associated with protecting the environment [66]. Many green behaviors, such as reducing car usage, conserving energy, and opting for organic food, necessitate individuals to exercise self-restraint for the betterment of the environment [67]. Consumers frequently encounter situations where they must make a choice between immediate personal advantages and long-term environmental benefits, such as a cleaner environment. Consequently, engaging in "going green" can be considered a virtuous behavior, as it requires consumers to forego personal benefits [68]. Nevertheless, engaging in acts of benevolence can also yield a hedonic reward by satisfying one's own sense of gratification [41].

Lindenberg and Steg [69] argued that the cultivation of pro-environmental attitudes and behaviors can be facilitated by the utilization of hedonic goal frames. Individuals are more inclined to embrace green behavior when they derive pleasure and satisfaction from engaging in pro-environmental actions [70]. Individuals consider green behaviors to be commendable pursuits due to the sense of personal satisfaction that arises from engaging in such behaviors [71]. Consumers who possess a higher level of hedonic shopping values exhibit greater responsiveness toward green issues and products compared with individuals with a lower level of hedonic shopping values.

Moreover, there is an association between legal wood and its price premium [15,16], resulting in a perception of luxury. Consequently, consumers who engage in hedonic consumption, seeking pleasure and gratification, are more likely to prefer legal wood. Therefore, we posit that:

**H2:** *Hedonic values positively relate to perception of green values toward legal wood.*

### 3.4. Interaction between Utilitarian and Hedonic Values

It is important to know that utilitarian (UV) and hedonic values (HV) are embedded within an individual and often interact with one another to shape an individual's perception and behavior. Therefore, once they are in play, several combinations of interactions may occur and result in various outcomes of perceptions or behaviors. Babin et al. [41] proposed that HV and UV are not mutually exclusive. In essence, it is possible for a consumer to concurrently possess multiple shopping values. For instance, when it comes to food consumption, humans engage in the actions not only for its fundamental purpose of sustenance, but also for the other aspects it encompasses. These aspects can include the food ingredients, serving experience, or social values. Naturally, this assertion is predicated on the condition that these people have their basic needs met. Inclusive considerations may also arise in the context of consumer perceptions of green products, particularly when utilitarian and hedonic values come into play. When individuals' utilitarian and hedonic values interact, there are two conceivable scenarios: these values are either aligned or in contradiction with each other.

Congruency occurs when the amounts of both utilitarian and hedonic values are equivalent: low-low, medium-medium, or high-high. In cases of congruency, individuals do not suffer any psychological conflict [72]. Consequently, the degree to which individuals see, hold attitudes toward, or exhibit behaviors relating to a certain entity will exhibit a positive correlation with the values they have engaged with. On the contrary, when these values are in disagreement, it creates a psychological conflict. When individuals encounter this conflict toward specific circumstances, they tend to exhibit a negative correlation between their attitudes or behaviors [72,73]. To illustrate, Kazen and Kuhl [73] demonstrated that when there is alignment between an employee's anticipated compensation and their actual salary, there is a notable increase in their job satisfaction.

In the present context of UV and HV, it may be observed that these two values inherently exhibit contrasting characteristics. Hence, the alignment between these factors will inevitably give rise to a psychological conflict among individuals. In discrepancy, individuals who have a higher preference for hedonic experiences over utilitarian ones, or vice versa, are likely to experience reduced psychological discomfort. Connecting these arguments within our legal wood context, we propose that:

**H3:** *When in discrepancy, if individuals' utilitarian values dominate over their hedonic values, and their perception of green values toward legal wood tends to be lower.*

**H4:** *When in discrepancy, if individuals' hedonic values dominate over their utilitarian values, their perception of green values toward legal wood tends to be higher.*

**H5:** *However, when utilitarian and hedonic values are in congruence, individuals' perception of green values toward legal wood tends to be lower.*

## 4. Materials and Methods

### 4.1. Samples and Procedures

Data collection was carried out in Semarang and Salatiga, two urban areas situated in the Central Java Province of Indonesia. Semarang serves as the administrative and economic center of Central Java Province, holding the role of both the capital and largest city within the region. Salatiga, being in close proximity to Semarang, has historically been closely linked to its economic progress. However, despite being located within the same agglomeration, the two cities exhibit distinct approaches to environmental regulation and green values. Therefore, the inclusion of data from these two cities provides the benefit of incorporating diverse contexts for comprehending green perception values on a broader level.

Respondents were residents of either city. They were recruited by open invitation disseminated via popular social media platforms in Indonesia, including WhatsApp groups, Facebook, and Instagram. This measure guaranteed the accessibility of our survey to a diverse range of consumers. We exclusively included respondents who were 17 years of age or older in order to guarantee their autonomy and consciousness regarding the survey questionnaire.

The survey was conducted using the online platform, Qualtrics. The questionnaire was also back translated into Indonesian by certified translator to ensure validity of the measurement. A study approval on human subjects was obtained from a review board in Universitas Brawijaya. In order to mitigate the potential influence of common method bias [74], two procedures were implemented during the data collection phase. First, participants were presented with a consent form at the commencement of the questionnaire, which outlined the requirements, potential outcomes, preservation of anonymity, and incentives associated with completing the survey. Additionally, participants were informed of their entitlement to retract their responses upon concluding the survey. Second, the participants were explicitly informed that there were no definitive or incorrect responses.

The data collection period was restricted to a duration of two weeks in August 2022, due to the recognition that our study encompasses a dynamic perception. Consequently, it was deemed crucial to employ a relatively short-term survey in order to effectively capture the phenomena under investigation, as opposed to utilizing a longer time frame. The inclusion of two cities that were implementing distinct environmental policies provided an opportunity to mitigate sample selection bias. The mean duration of survey questionnaire completion was 10 min. In exchange for participating in the survey, respondents were given the opportunity to enter a randomized draw for the chance to win e-money vouchers.

A total of 731 responses were collected; however, only 515 of these responses met the eligibility criteria due to their alignment with legal wood-related questions and completeness. In total, our respondents were 54.3% male and 45.7% female, and 85.1% of them had obtained a higher education level with an average age of 40.14 years old (see Table 1). We possess a high level of confidence in the fairness and representativeness of this sample in relation to the population. In order to ensure that the content validity of each statement was pertinent to individuals from Indonesia, the survey questionnaire underwent translation by a certified translator. The translations were subsequently subjected to a rigorous verification process conducted by specialists in the fields of management science and forest/wood science.

**Table 1.** Demographic characteristics.

|  |  | Frequency | Percentage |
|---|---|---|---|
| Age | <31 | 110 | 21.4 |
|  | 31–40 | 146 | 28.4 |
|  | 41–50 | 169 | 32.8 |
|  | 51–60 | 86 | 16.7 |
|  | >60 | 4 | 0.7 |
| Education | Low and Middle | 77 | 14.9 |
|  | Higher | 438 | 85.1 |
| Gender | Men | 280 | 54.3 |
|  | Women | 235 | 45.7 |

*4.2. Measures and Analysis*

All survey constructs were measured using multiple items from prior literature (Appendix A). Hedonic and utilitarian values in shopping behavior were measured with items from Cheng et al. [23] and Babin et al. [41], revised to suit our research context. The utilitarian values (UV) scale contained six items (M = 3.90, SD = 0.66, $\alpha$ = 0.78), and the hedonic values (HV) scale comprised eleven questions (M = 3.08, SD = 0.65, $\alpha$ = 0.85). Perception of green values (PGV) was measured with five items adapted from Chen [75] (M = 4.10, SD = 0.61, $\alpha$ = 0.90). We also included three control variables (age, level of education, and gender) to have a better understanding of the true relationship between shopping values and perception of green values, as older generations, higher educated persons [76], and women [77] tend to have greater awareness of the environment.

This study investigates not only the linear relationship but also the effect of discrepancy and congruency between HV and UV, and how that affects perception of green values toward legal wood. The utilization of polynomial regression combined with surface analysis was deemed appropriate for examining the aforementioned effects [78]. The utilization of this analytical approach enables the comprehension of intricate disparities and agreements among autonomous variables, as well as their directional association with the reliant variable [79]. Harris et al. [80] and Kazén and Kuhl [73] provide comprehensive reviews on polynomial regressions with surface analysis, offering valuable insights into the application and interpretation of this methodology. These works serve as excellent resources for understanding the intricacies of polynomial regressions and offer practical guidelines for utilizing and making sense of the results obtained using this approach.

**5. Results**

Table 1 shows that our respondents were relatively distributed among productive ages and genders. We believe that this composition is representative of the population of the study, although they were skewed toward having a higher education background (having at least a college education).

Table 2 shows the correlations between the study variables. The results show that, in general, the level of HV of our respondents was considered medium compared with the level of UV. However, the level of perception of green values was indicated as high. We also found a negative correlation between individuals' perceptions of green values (PGV) and UV, and a positive correlation between PGV and HV. From the correlation analysis, we found that female respondents had significantly lower levels of PGV and UV, but higher levels of HV. It was also interesting to know that the higher education level of our respondents corresponded to a lower level of HV. We then explored these relationships in a series of hierarchical Ordinary Least Squares (OLS) regressions, followed by a polynomial regression with surface analysis.

**Table 2.** Descriptives and correlations of the main and control variables.

| | | Mean | SD | 1 | 2 | 3 | 4 | 5 | 6 |
|---|---|---|---|---|---|---|---|---|---|
| 1 | Education | 1.8 | 0.35 | 1 | | | | | |
| 2 | Gender | 1.46 | 0.99 | −0.002 | 1 | | | | |
| 3 | Age | 40.14 | 10.13 | 0.317 ** | −0.108 ** | 1 | | | |
| 4 | PGV | 4.17 | 0.67 | 0.031 | −0.132 ** | 0.093 ** | 1 | | |
| 5 | UV | 3.90 | 0.66 | −0.055 | −0.224 ** | 0.055 | −0.258 ** | 1 | |
| 6 | HV | 3.08 | 0.65 | −0.081 ** | 0.104 ** | −0.183 ** | 0.105 ** | 0.304 ** | 1 |

** $p < 0.05$, HV: hedonic values, UV: utilitarian values, PGV: perception of green values.

Results from hierarchical OLS and polynomial regressions are shown in Table 3. The hierarchical OLS showed better R2 and adjusted R2 along four model fits. This indicates that the studied variables and their relationships with one another were capable of explaining the phenomenon. An interesting pattern was seen in Model 2, where we failed to prove that, independently, utilitarian values had a negative relationship with perception of green values toward legal wood. In Model 2, UV had a positive relationship with PGV. Thus, our H1 was not supported. However, we also found that, independently, hedonic value had a positive relationship with perception of green values toward legal wood. Therefore, our H2 was supported.

We also tested the effect of the interaction between UV and HV toward perception of green values. In Model 3, we found that the interaction had a significant and negative relationship on perceptions of green values. This was the first indication that there were possible incongruency effects on this relationship. Hence, we performed a simple slope analysis to confirm this possibility. We found that when HV interacts with UV, the positive effects are only significant at low (effect = 0.399, SE = 057, t = 6.917, LLCI = 0.280 and ULCI = 0.512) and medium (effect = 0.235, SE = 0.040, t = 5.834, LLCI = 0.156 and ULCI = 0.315) levels of utilitarian values. When it came to high levels of UV, the effect was no longer significant. To visualize this relationship, we provide a graphical illustration of the simple slope analysis in Figure 5.

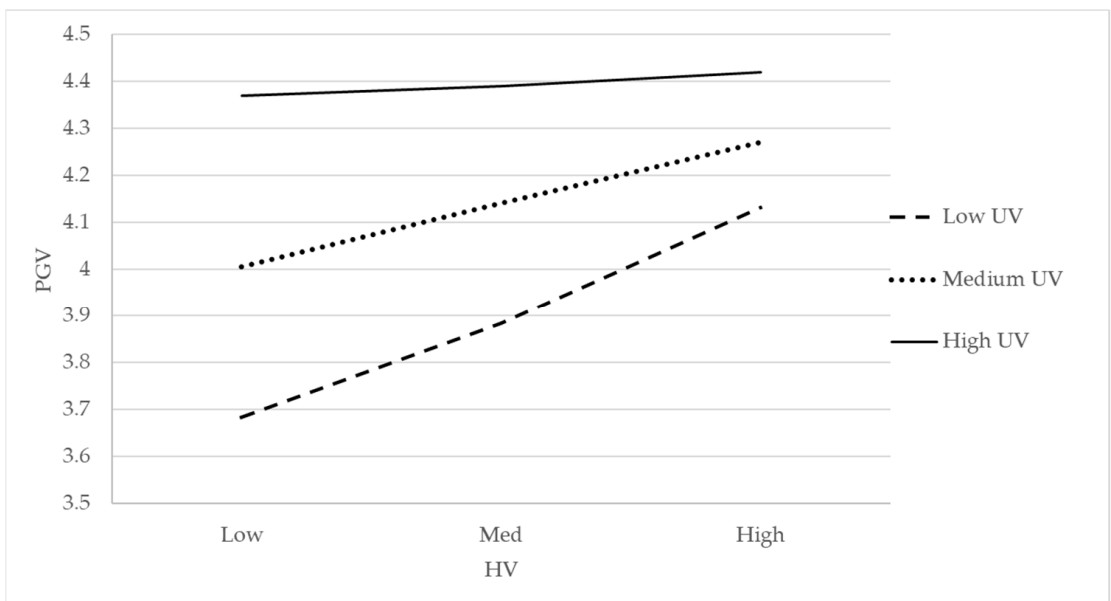

**Figure 5.** Graphic result of interaction of utilitarian values (UV) and hedonic values (HV) toward perception of green values (PGV).

**Table 3.** Hierarchical OLS and polynomial regression of the main and control variables.

| Variables | Model 1 | | | Model 2 | | | Model 3 | | | Model 4 | | |
|---|---|---|---|---|---|---|---|---|---|---|---|---|
| | B | SE | VIF | B | SE | VIF | B | SE | VIF | B | SE | VIF |
| Control Variables | | | | | | | | | | | | |
| Education | −0.001 | 0.020 | 1.140 | 0.010 | 0.019 | 1.145 | 0.006 | 0.019 | 1.148 | 0.042 | 0.074 | 1.157 |
| Gender | −0.152 ** | 0.055 | 1.019 | −0.085 | 0.052 | 1.070 | −0.076 | 0.051 | 1.071 | −0.069 | 0.051 | 1.124 |
| Age | 0.005 * | 0.003 | 1.151 | 0.006 ** | 0.003 | 1.181 | 0.006 ** | 0.003 | 1.181 | 0.006 ** | 0.003 | 1.070 |
| Constant | 4.205 | 0.183 | | 2.089 | 0.305 | | −1.188 | 0.773 | | 1.789 | 1.269 | |
| Main Variables | | | | | | | | | | | | |
| UV | | | | 0.319 ** | 0.041 | 1.124 | 1.131 ** | 0.181 | 23.35 | −0.282 | 0.435 | 137.1 |
| HV | | | | 0.208 ** | 0.041 | 1.107 | 1.204 ** | 0.221 | 33.73 | 0.97 ** | 0.394 | 110.1 |
| HV and UV Interaction | | | | | | | −0.248 ** | 0.054 | 41.98 | −0.183 ** | 0.058 | 51.4 |
| UV$^2$ | | | | | | | | | | 0.161 ** | 0.045 | 83.0 |
| HV$^2$ | | | | | | | | | | −0.006 | 0.034 | 49.4 |
| Model Fit | | | | | | | | | | | | |
| .sig | | 0.008 | | | 0.000 | | | 0.000 | | | 0.000 | |
| F | | 4.143 | | | 17.453 | | | 18.615 | | | 15.885 | |
| R Square | | 0.024 | | | 0.146 | | | 0.180 | | | 0.200 | |
| Adj R Square | | 0.018 | | | 0.138 | | | 0.171 | | | 0.188 | |
| $\Delta R^2$ Model 1 to 2 | | | | | | | | | | | | |
| $\Delta R^2$ Model 2 to 3 | | | | | 0.104 | | | | | | | |
| $\Delta R^2$ Model 3 to 4 | | | | | | | | 0.034 | | | | |
| $\Delta R^2$ Model 4 to 5 | | | | | | | | | | | 0.020 | |

HV: hedonic values, UV: utilitarian values, Model 4 is the polynomial regression. * $p < 0.10$, ** $p < 0.05$.

Further, to test the congruencies and discrepancies of the interaction between UV and HV, we performed a polynomial regression with surface analysis (Table 4). The results of our polynomial regression showed that the interaction was indeed consistent with the OLS regression. Next, we confirmed the surface analysis in Table 4, and found that when in congruence, the interaction between UV and HV had no significant effect on PGV toward legal wood. Thus, our H5 was not supported.

**Table 4.** Testing result of surface analysis.

| | | Standard | Test | |
|---|---|---|---|---|
| **Effect** | **Coefficient** | **Error** | **Stat (t)** | ***p*-Value** |
| a1: Slope along x = y (as related to Z) | 0.69 | 0.70 | 0.982 | 0.326 |
| a2: Curvature on x = y (as related to Z) | −0.03 | 0.11 | −0.264 | 0.792 |
| a3: Slope along x = −y (as related to Z) | 1.25 | 0.45 | 2.810 | 0.005 |
| a4: Curvature on x = −y (as related to Z) | 0.34 | 0.09 | 3.973 | 0.000 |

Meanwhile, when there was a discrepancy between values, with UV being higher than HV or vice versa, the effects were significant both in slope and curvature shapes (Figure 6). Therefore, our H3 and H4 were supported. In sum, we prove that, in the context of legal wood, the perception of green values of consumers is higher when their HV is dominant.

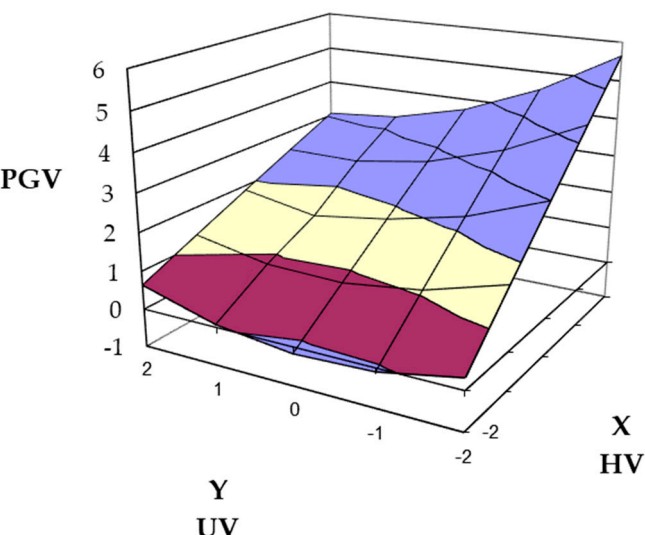

**Figure 6.** Graphical result of the surface analysis: PGV on HV and UV.

## 6. Discussion and Conclusions

Our data analysis supported Hypotheses 2, 3, and 4, but not Hypotheses 1 and 5. Positive green perceptions toward legal wood were associated with hedonic shopping values, which aligned with conclusions drawn from other studies, suggesting that self-indulgence and pleasure can positively influence green behaviors [69,81]. Conversely, possessing utilitarian shopping values diminished an individual's level of environmental engagement. Utilitarian shopping values, as suggested by the CLT [59,60], directs consumers' attention toward the tangible advantages offered by a product or service. This results in a reduced amount of cognitive processing. Thus, individuals are consequently less inclined to show concern for the environment, as being green with a preference for sustainable

products/choices is associated with a higher level of construal, meaning they are more distant and abstract in nature [40].

The outcomes of our research have significant importance for wood marketers. Hedonic and utilitarian shopping values can be understood as the result of a complex interaction that encompasses not only the products themselves, but also the consumers and distribution channels [82–85]. From a pragmatic standpoint, our findings indicate that wood marketers could benefit from directing their communication efforts toward emphasizing the hedonic worth of the product, as opposed to its utilitarian values. It is imperative for distributors and promoters of green wood products to carefully deliberate on strategies to effectively elicit consumers' hedonic shopping values that are inherently linked to the utilization of green products. To enhance hedonic shopping values, marketers should place emphasis on the hedonic characteristics of different product qualities, such as packaging, style, and product display. Marketers have the option to engage in communication with consumers through the utilization of emotional appeals, as such appeals align effectively with hedonic shopping values [86]. This can also be accomplished utilizing digital marketing applications on smartphones [87].

When legal wood is associated with certain brands, it is possible to establish a connection between the brand and emotional advantages, through strategic marketing and advertising efforts [88]. Companies who operate within the environmentally conscious sector or are engaged in the repositioning of green products or brands should aim to augment consumers' heightened perceptions of control with respect to environmental matters and associated green consumption. These actions should also be supported by public policy marketers, such as administrators of educational forums, awareness campaigners, and non-profit organizations [89]. They have a responsibility to prioritize the enhancement of individuals' sense of personal control in relation to environmental results. Enhancing the individual's perceived influence over environmental issues is of utmost importance.

*Future Directions and Study Limitations*

Our results also lead to the question of whether our respondents, or Indonesian consumers in general, are aware about using legal wood products. To illustrate, some customers are loyal to IKEA furniture without knowing that the company has pledged to ensure their supply chains are free of illegally sourced wood. Indonesian customers are probably attracted to IKEA's neuromarketing strategies, such as their attractive displays and labyrinth effect, without realizing that IKEA also applies green marketing and supports green products. As mentioned on their website, IKEA has implemented a thorough, due diligence system to manage their wood supply [90]. The system encompasses many measures, including the ability to track the origin of the wood utilized in their products.

The arguments stated above require further investigation. In addition to wood-based furniture, further study should be directed to what Indonesians believe about legal wood utilization in construction [91].

Though this study has both theoretical and practical consequences, we acknowledge certain limitations. First, the participants in this empirical research were relatively well-educated, and the limited size of the sample places limits on generalizing the findings to individuals with lesser educational backgrounds. Second, another constraint to consider is the degree of honesty in the responses offered. Due to the inherent characteristics of the subject matter under investigation, namely environmental attitudes, it was difficult to mitigate the tendency of respondents to exclusively furnish socially favorable responses. Third, an additional limitation pertains to the cross-sectional and correlational design employed in this study. Though personality theorists acknowledge that psychological traits can cause differences in behavior [92], it is important to note that the existing research does not offer conclusive evidence of these causal links.

**Author Contributions:** Conceptualization, P.L. and R.P.H.; methodology, P.L. and T.F.; software, R.P.H. and T.F.; validation, P.L. and R.P.H.; formal analysis, R.P.H.; investigation, P.L.; resources, P.L., R.P.H., T.F. and D.M.S.; data curation, R.P.H. and T.F.; writing—original draft preparation, P.L.; writing—review and editing, P.L. and R.P.H.; visualization, R.P.H.; supervision, P.L.; project administration, R.P.H. and D.M.S.; funding acquisition, P.L. and R.P.H. All authors have read and agreed to the published version of the manuscript.

**Funding:** This study is funded by the Faculty of Economics and Business of Universitas Brawijaya, contract number 5285.2/UN10.F02/TU/2022. The open access is supported by the Arkansas Center for Forest Business.

**Data Availability Statement:** Data is available upon request. Please contact radityohandrito@ub.ac.id for more information.

**Conflicts of Interest:** The authors declare no conflict of interest.

## Appendix A

**Table A1.** Survey questionnaire.

| Utilitarian Values | Hedonic Values | Perception of Green Values toward Legal Wood |
|---|---|---|
| I accomplish just what I want to on a shopping trip | Going shopping is truly a joy | The environmental functions of legal wood provide very good value for me |
| I am disappointed because I have to to go to another store(s) to complete my shopping | I continue to shop, not because I have to, but because I want to | The environmental performance of legal wood meets my expectations |
| A good store visit to me is one that is quick | Compared to other things I could have done, the time spent shopping is truly enjoyable | I use legal wood because it has more environmental concern (than other products) |
| While shopping, I find just the item(s) I am looking for | I enjoy a shopping trip for its own sake, not just for the items I may have purchased | I use legal wood because it is environmentally friendly |
| I feel smart about my shopping decisions | I have a good time during a shopping trip because I am able to act on the "spur of the moment" | I use legal wood because this product has positive benefits for the environment. |
| I can buy what I really need | While shopping, I am able to forget my problems | |
| | During a shopping trip, I feel the excitement of the hunt | |
| | Going shopping is not a very nice time out (reversed) | |
| | Going shopping truly feels like an escape | |
| | While shopping, I feel a sense of adventure | |
| | I enjoy being immersed in exciting new products | |

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
