# Peer review of "Who Prefers Legal Wood: Consumers with Utilitarian or Hedonic Shopping Values?"

_forests, doi:10.3390/f14112163_

Round 1
Reviewer 1 Report
Comments and Suggestions for Authors
Dear Authors,
Thank you for putting effort to come out this manuscript. Please the attached for minor suggestions.
Reviewer

Author Response
Thank you for your comments and suggestions. Please see the attachment for our detailed responses.

Reviewer 2 Report
Comments and Suggestions for Authors
Dear Editor-in-Chief,
The manuscript by Larasatie et al. is on consumer preferences for legal wood, econpassing a comparison between utilitarian and hedonic shoppers. Overall, the study is well-conducted and provides valuable insights into consumer preferences for legal wood. The manuscript is well-structured, and the introduction, methodology, results, and conclusions are well-developed. However, there are a few suggestions and concerns that should be addressed before publication.
Strengths:
- The introduction provides relevant background information and highlights the significance of the study.
- The methodology is well-described, allowing for the study's reproducibility by readers with a strong background.
- The results are presented in an engaging manner, with practical implications discussed for each table or graph.
- The conclusions are closely tied to the results, acknowledge the limitations of the methodology, and effectively summarize the key messages of the article.
Suggestions for Improvement:
- While the overall textual quality of the manuscript is good, there are scattered grammatical errors throughout the text. I recommend using grammar-checking software, even free options, to improve the text's clarity and correctness.
- Figure 1 exhibits low aesthetic quality. It would be preferable to use a larger font and enhance the figure's plot area. Placing the legend within the plot area could improve the figure's overall presentation.
- In the introduction, it would be beneficial if the authors could incorporate statistical data related to the Indonesian forest and timber market. This addition would provide context and enhance the understanding of the study's relevance.
- In the methodology section, consider including more information about the questionnaire used. Providing a supplementary material with the list of questions answered by the respondents could enhance transparency and facilitate a more comprehensive understanding of the research methods employed.
Addressing the above-mentioned suggestions would further enhance the quality of the article. I recommend that the manuscript undergo minor revisions to address these points before being considered for publication in the Forests journal.
Thank you for considering my review, and I look forward to seeing the revised manuscript.
Sincerely,
Comments on the Quality of English LanguageDear Editor-in-Chief,
The manuscript by Larasatie et al. is on consumer preferences for legal wood, econpassing a comparison between utilitarian and hedonic shoppers. Overall, the study is well-conducted and provides valuable insights into consumer preferences for legal wood. The manuscript is well-structured, and the introduction, methodology, results, and conclusions are well-developed. However, there are a few suggestions and concerns that should be addressed before publication.
Strengths:
- The introduction provides relevant background information and highlights the significance of the study.
- The methodology is well-described, allowing for the study's reproducibility by readers with a strong background.
- The results are presented in an engaging manner, with practical implications discussed for each table or graph.
- The conclusions are closely tied to the results, acknowledge the limitations of the methodology, and effectively summarize the key messages of the article.
Suggestions for Improvement:
- While the overall textual quality of the manuscript is good, there are scattered grammatical errors throughout the text. I recommend using grammar-checking software, even free options, to improve the text's clarity and correctness.
- Figure 1 exhibits low aesthetic quality. It would be preferable to use a larger font and enhance the figure's plot area. Placing the legend within the plot area could improve the figure's overall presentation.
- In the introduction, it would be beneficial if the authors could incorporate statistical data related to the Indonesian forest and timber market. This addition would provide context and enhance the understanding of the study's relevance.
- In the methodology section, consider including more information about the questionnaire used. Providing a supplementary material with the list of questions answered by the respondents could enhance transparency and facilitate a more comprehensive understanding of the research methods employed.
Addressing the above-mentioned suggestions would further enhance the quality of the article. I recommend that the manuscript undergo minor revisions to address these points before being considered for publication in the Forests journal.
Thank you for considering my review, and I look forward to seeing the revised manuscript.
Sincerely,
Reviewer 3 Report
Comments and Suggestions for Authors
Dear manuscript authors
Your study under the theme "Who prefers legal wood: Consumers with utilitarian or hedonic shopping values?" seems to be interesting and useful. However, I have doubts about the quality and acceptability of its publication in the form of an article.
Firstly, I am confused by the quality of the questionnaire you developed to collect information. If the survey was sent to respondents as presented in Appendix A, its quality is unfortunately low.
The questions are confusing and difficult to understand. The text of the manuscript does not mention whether the first version of the questionnaire was piloted to gather feedback on improvements and then the final version was sent to participants. By what criteria were survey participants selected and was the anonymity of their names guaranteed? Why was the survey not designed for buyers with varying purchasing power?
Forest resource management and certification is part of my research in Europe, but the lack of description of the Indonesian situation in the introduction to your manuscript limited my attempts to better understand the subject of the study.
Numerous hypotheses are confusing. It is better to transform this part into the form of research objectives and define 1-2 hypotheses.
The "Methodology" part lacks of the information about the development and implementation of the survey, as well as the methods for analyzing the results. For example, it would be interesting to know whether there is a relationship between the choice of answers and the gender of the respondents or their age.
More detailed questions are described below:
The introduction to this article lacks information/ description of Indonesia's forests and their management. There is no description of the size of the forest areas, type of ownership, or actors involved in their management. Therefore, it is difficult to understand the study area.
Who are the stakeholders and why are neighbours and communities important in the governance of Indonesia's forests?
It is not clear what national certification is in the introductory part of the paper. Next comes criticism of the consequences of certification, but it is not clear whether this criticism applies to international certification systems (for example FSC) or national systems.
156-161 - Equating the consumption of food during famine with green products appears to be strange/incorrect. The logic of the comparison is poorly explained by the authors of this article.
162-177 - there are no references.
178-185 The text is written in a confusing way. It is not clear whether people with utilitarian values are for or against green products.
251-254 It's a confusing sentence, and it's not clear why reference 65 is included.
254 Why is the example of the mentioned correlation not presented?
259-262 These sentences are opposite to sentences 251-252. It is recommended to rewrite this part to make the authors' idea clearer.
H1 is opposite H2 and H3 is opposite H4. I recommend understanding better the concept of Hypothesis and how it is usually established for scientific research. There is no need to write opposing hypotheses. Just write one and then, based on the study and detected arguments, accept or reject this
299 Replace Methods with Materials and methods
3.1. Samples and Procedure.
How were the stakeholders chosen for the study?
How many questions did the survey include and how were they organized?
Is there information on the number of people the survey was sent to? This will give us an idea of the % of those who received an invitation and responded.
Based on what criteria do the authors consider the 731 responses as a representative number on the national scale?
332-341 this information is already part of the research results.
332-334 30% of the responses were incomplete. This was despite the fact that the majority of those interviewed had higher levels of education. If the questionnaire is the same as in Appendix A, a loss is understandable. In addition, the complexity of the survey questions excluded the possibility of collecting the opinions of people with lower levels of education, which is incorrect and does not respect the rights of this type of consumer to express their opinions. It is recommended that authors adopt a more professional approach to preparing such types of questionnaires in the future.
3.2. Measures and Analysis - The number of types of analysis used is scarce and probably does not allow for a complete analysis of the data obtained.
4. Results
Table 1 – The results regarding education are not clear. Information about only 15% of respondents. What kind of education do the other participants have? Or was this question not required?
Table 2 - The table name is incomplete. It is not clear what the numbers with two stars mean.
Table 3 - Table legend is uninformative. The 4 models and abbreviations must be explained, as well as just like abbreviations
It would be interesting to know whether there is a relationship between the choice of answers and the gender or age of the respondents. I also think that similar comparisons with other variables could possibly reveal something interesting.
4. Discussion and Conclusions - There is an error in the numbering of this section
421-422 - Judging by the descriptions of Hypotheses 3 and 4, they are opposite. It seems strange how these two hypotheses were confirmed by analyzing the data obtained in the survey.
423-424 The description talks about other studies, but only provides one reference.
428-430 The sentence is very confusing and the logic of reflecting on the color green is not clear.
446 -465 This text does not include references to other studies, therefore it cannot be considered a discussion.
Overall, based on the results obtained, the discussion could have been more interesting.
The last two conclusions of this study are unclear and require a deeper explanation.
Thank you for your attention and wish you success with the next version of the document.
Comments on the Quality of English Language
Please check the document's grammar.
Reviewer 4 Report
Comments and Suggestions for Authors
Dear Authors
The study’s importance from an end-user perspective lies in its potential to inform and empower consumers who care about the legality, ethics, and sustainability of the wood products they purchase.
The manuscript presents a new data processing method that provides accurate information about these resources that can help them make more informed choices, advocate for change, and have a positive impact on the wood industry and the environment which is crucial in achieving the goals above.
The work is invaluable and both academic and practical relevance can be found in the problem choice and importance.
Below are suggestions for improvement of the manuscript
Point 1 |
The research question is not explicitly stated in the introduction hence Scope to improve research gaps and research contribution |
Point 2 |
Scope to improve the value of the study |
Point 3 |
Scope to improve the practical implication of the results for consumers, businesses and policymakers, hence there is scope to improve it |
Point 4 |
Please explain the limitations of the study, and future research direction more clearly. |
Round 2
Reviewer 3 Report
Comments and Suggestions for Authors
This article “Who prefers legal wood: Consumers with utilitarian or hedonic shopping values?” - is interesting and relevant. I recommend accepting the manuscript for publication after minor corrections:
Correct the legend in Figure 2. The names of some countries are not visible, and abbreviations may be used, for example in the case of USA.
65-66 – grammar mistake. The reviewer only asked that the word “mentioned” be deleted, but that the word “strikethrough” not be added to the text. Please check the meaning of this last term in the dictionary.
105 - They have the belief believe that
413-417 - This information belongs to the “Results” section.
Comments on the Quality of English Language
Check and correct English of document.
